

# A hybrid model of modal decomposition and gated recurrent units for short-term load forecasting

Chun-Hua Wang[1] and Wei-Qin Li[2]

[1] School of Electronic Engineering, Xi'an Aeronautical Institute, Xi'an, Shaanxi, China
[2] School of Automation and Information Engineering, Xi'an University of Technology, Xi'an, Shaanxi, China

## ABSTRACT

Electrical load forecasting is important to ensuring power systems are operated both economically and safely. However, accurately forecasting load is difficult because of variability and frequency aliasing. To eliminate frequency aliasing, some methods set parameters that depend on experiences. The present study proposes an adaptive hybrid model of modal decomposition and gated recurrent units (GRU) to reduce frequency aliasing and series randomness. This model uses average sample entropy and mutual correlation to jointly determine the modal number in the decomposition. Random adjustment parameters were introduced to the Adam algorithm to improve training speed. To assess the applicability and accuracy of the proposed hybrid model, it was compared with some state of the art forecasting methods. The results, which were validated by actual data sets from Shaanxi province, China, show that the proposed model had a higher accuracy and better reliability compared to the other forecasting methods.

## INTRODUCTION

Electrical load forecasting plays an important role in power system dispatching and security detection. However, it is very difficult to accurately forecast load because the electrical load can be affected by the weather and some accidental factors (*Mideksa & Kallbekken, 2010*; *Wang et al., 2012*). It is, therefore, important to develop an effective load forecasting method that is both reliable and accurate.

Electrical load measurement data can be contaminated by random noise that reduces accurate forecasting performance (*Xiao et al., 2007*; *Li, 2020*; *Ren & Li, 2023*). Signal processing technologies have been developed to reduce the random noise created during measurement, such as faults in the sensors or power supply equipment failures (*Guan et al., 2021*; *Wang, Yao & Papaethymiou, 2023*). Filtering methods, such as the wavelet analysis method and the Kalman filtering method, are currently the most common ways of dealing with the random noise in the data (*Quilty & Aadamowski, 2018*; *Nobrega & Oliveira, 2019*). Electrical load is also affected by people's consumption habits and varies drastically between

Corresponding author
Wei-Qin Li, wqlee@xaut.edu.cn

different periods of time, so frequency aliasing, which occurs when the load sequence is not sampled at a high enough rate, is a significant problem and makes it harder to accurately forecast load from the data. Some methods, including empirical mode decomposition (EMD) and variational mode decomposition (VMD), reduce aliasing by decomposing the load series (*Li & Chang, 2018*; *Mounir Nada, Ouadi & Jrhilifa, 2023*; *Rayi et al., 2022*). The disadvantage of these methods is that they use parameters that depend on experiences.

Short-term load forecasting techniques include statistical models (*Ren & Li, 2023*; *Lee & Ko, 2021*; *Jin et al., 2021*), the machine learning method (*Tarmanini et al., 2023*; *Xie et al., 2020*) and the deep learning method. In these methods, the long and short-term memory (LSTM) network can find the evolution characteristics of a time series based on a large number of training samples, resulting in a higher accuracy than traditional machine learning methods (*Mokarram et al., 2023*; *Rafi et al., 2021*). However, LSTM training time is long, the structure of the LSTM network is complex, and its parameters are difficult to determine. Compared with the LSTM network, the gated recurrent unit (GRU) network developed in recent years has a simpler structure and reduces computational complexity, so it has also been applied in time series prediction (*Jung et al., 2021*; *Pu et al., 2023*; *Li et al., 2023*).

This article focuses on a hybrid model of adaptive VMD and the GRU network for short-term load forecasting. To reduce aliasing and random noise, this model uses an adaptive VMD method, determining the modal number with the average sample entropy and mutual correlation. This model also uses the GRU network, and further reduces training time by expanding the random adjustment parameters of the Adam algorithm. The electric load forecasting results of this proposed model were compared with other state of the art forecasting methods and found to be both accurate and reliable.

## ADAPTIVE VMD FOR LOAD SERIES

This section first introduces the basic principles of VMD, and then proposes an adaptive VMD model to reduce aliasing and random noise in the load series.

### Variational mode decomposition

VMD decomposes the original series into several intrinsic mode functions (IMFs), each of which is a sub-sequence of the frequency modulation and amplitude modulation (*Dragomiretskiy & Zosso, 2014*; *Zhang & Guo, 2020*). VMD demodulates the IMF to its own fundamental frequency bandwidth and aims to minimize total modal bandwidth to find the optimal IMF. The VMD includes both variational construction and a variational solution.

To obtain the analytical signals of each IMF and the corresponding unilateral spectrum, The modal function obtained by decomposition, represented as: $u_k(t)$,k=1 ,2,…,$K$ is processed with Hilbert transform (*Huang et al., 1998*) as follows:

$$\left(\delta(t)+\frac{j}{\pi t}\right)*u_k(t) \tag{1}$$

where $\delta(t)$ is the unit impulse signal.

Center frequency, represented as $\omega_k$, is then multiplied by the exponential term $e^{-j\omega_k t}$ to modulate the spectrum of the mode to its fundamental frequency:

$$\left[\left(\delta(t)+\frac{j}{\pi t}\right)*u_k(t)\right]e^{-j\omega_k t} \tag{2}$$

The bandwidth of the mode is then calculated, using the solution of the 2-Norm of the modulated signal gradient to solve the variational constraint problem:

$$\min_{u_k,\omega_k}\left\{\sum_{k=1}^{K}\left\|\partial_t\left[\left(\delta(t)+\frac{j}{\pi t}\right)u_k(t)\right]e^{-j\omega_k t}\right\|_2^2\right\}$$
$$s.t.\sum_{k=1}^{K}u_k(t)=f(t) \tag{3}$$

where $f(t)$ is the original signal.

The variational constraint problem is then transformed into an unconstrained problem using the Lagrange multiplier method and quadratic multiplication operator alternation algorithm. By introducing the Lagrange multiplication operator $\theta(t)$ and penalty factor $C$ into the constraint problem, the unconstrained problem is as follows:

$$L(u_k,\omega_k,\theta)=C\sum_{k=1}^{K}\left\|\partial_t\left[\left(\delta(t)+\frac{j}{\pi t}\right)u_k(t)\right]e^{-j\omega_k t}\right\|_2^2+\left\|f(t)-\sum_{k=1}^{K}u_k(t)\right\|_2^2$$
$$+\left\langle\theta(t),f(t)-\sum_{k=1}^{K}u_k(t)\right\rangle. \tag{4}$$

Then, the optimization problem of $u_k$ can be obtained by using the multiplication operator alternating algorithm:

$$u_k^{n+1}=\underset{u_k\in X}{\text{argmin}}\left\{\theta\left\|\partial_t\left[\left(\delta(t)+\frac{j}{\pi t}\right)u_k(t)\right]e^{-j\omega_k t}\right\|_2^2+\left\|f(t)-\sum_{k=1}^{K}u_k(t)\right\|_2^2\right.$$
$$\left.+\left\|f(t)-\sum_{i=1}^{K}u_i(t)+\frac{\theta(t)}{2}\right\|_2^2\right\} \tag{5}$$

where $i$ is the iteration control parameter. By using the Parseval/Plancherel Fourier equidistant method under 2-norm, it can be obtained as follows:

$$\hat{u}_k^{n+1}=\underset{\hat{u}_k,u_k\in X}{\text{argmin}}\left\{\int_0^{\infty}4\theta(\omega-\omega_k)^2|\hat{u}_k(\omega)|^2+2\left|\hat{f}(\omega)-\sum_{i=1}^{K}\hat{u}_i(\omega)+\frac{\hat{\theta}(\omega)}{2}\right|^2 d\omega\right\}. \tag{6}$$

Therefore, the optimized solution for this quadratic problem is:

$$\hat{u}_k^{n+1}(\omega)=\frac{\hat{f}(\omega)-\sum_{i\neq k,i=1}^{K}\hat{u}_i(\omega)+\frac{\hat{\theta}(\omega)}{2}}{1+2C(\omega-\omega_k)^2}. \tag{7}$$

Finally, the center frequency can calculated using the following quadratic formula:

$$\omega_k^{n+1}=\frac{\int_0^{\infty}\omega|\hat{u}_k(\omega)|^2 d\omega}{\int_0^{\infty}|\hat{u}_k(\omega)|^2 d\omega}. \tag{8}$$

## Adaptive VMD

As a non-recursive decomposition method, the number of modes in VMD needs to be set in advance. When the number of modes is too small, there is insufficient decomposition; also, some modal functions can result in false spectrum and spectrum breakage (*Jiang, Shen & Shi, 2018*). In our previous work, the number of modes is set according to the cross-correlation coefficient (*Shang, Li & Wu, 2023*). In the proposed adaptive VMD model, to improve reliability the number of modes is jointly determined by the cross-correlation coefficient and the average sample entropy.

### The correlation coefficient

The correlation coefficient reveals the correlation degree of two sequences; the higher the correlation between two sequences, the closer the cross-correlation coefficient is to 1 (*Shang, Li & Wu, 2023*). With the residual sequence represented as $l(n)$ and the modal function represented as $f(n)$, the standard cross-correlation coefficient $\rho_c$ is defined as:

$$\rho_c = \sum_{n=0}^{N} f(n)l(n) / \sqrt{R_{ff}R_{ll}} \tag{9}$$

where $N$ is the sequence length, and $R_{ll}$ and $R_{ff}$ are the autocorrelation coefficients of $l(n)$ and $f(n)$, respectively.

### The average sample entropy

Each mode after VMD decomposition has its own central frequencies, so the spectrum will not overlap. As a result, the similarity of each mode is high, and the sample entropy is small. When the number of decompositions is the optimal value, the sample entropy of each mode (except the residuals) and the average sample entropy (ASE) should both be the smallest (*Lake, 2010*; *Sun & Wang, 2018*).

With the time series represented as $Y(n), n=1,2,\ldots,N$, and the modal function represented as $u_k(n), k=1,2,\ldots,K$, $U(i)$ is obtained by the modal function of $u_k(i)$, as follows:

$$U(i) = [u_k(i), u_k(i+1), \ldots, u_k(i+m-1)] \tag{10}$$

where $i = 1, 2, \ldots, N-m+1$.

Firstly, the maximum distance $d_m[U(i), U(j)]$ between the corresponding elements of $U(i)$ and $U(j)$ is defined as:

$$d_m[U(i), U(j)] = \max_{l=0,1,\ldots,m-1} |u_k(i+l) - u_k(j+l)|. \tag{11}$$

Then, counting the number of $j$ satisfying the formula $d_m[U(i), U(j)] < r$ for each $i$, defined as $B_i$. Here, $j$ satisfies $N - m \geq j \geq 1$ and $r$ is the tolerance of similarity measure. Based on this, the ratio of $B_i^m$ to the total distance of $N-m$ is as follows:

$$B_i^m(r) = \frac{B_i}{N-m}. \tag{12}$$

Next, the average value $B^m(r)$ of $B_i^m(r)$ is calculated as follows:

$$B^m(r) = \sum_{i=1}^{N-m} \frac{B_i^m(r)}{N-m+1}. \tag{13}$$

Wang and Li (2023), *PeerJ Comput. Sci.*, DOI 10.7717/peerj-cs.1514

Lastly, by increasing the dimension to $m + 1$, the average value of $B_i^{m+1}$ is obtained as follows:

$$B^{m+1}(r) = \sum_{i=1}^{N-m} \frac{B_i^{m+1}(r)}{N-m} \tag{14}$$

where $B^m(r)$ and $B^{m+1}(r)$ are the probability that two sequences match $m$ and $m + 1$ points under the tolerance of similarity measure $r$, respectively. The sample entropy of the modal sequence can then be written as:

$$S_E = -\ln\left[\frac{B^{m+1}(r)}{B^m(r)}\right]. \tag{15}$$

Equation (15) shows that sample entropy is related to both $m$ and $r$. eference previous study (*Pincus, 2001*) showed that when $r$ is 1 or 2 and $m$ is 0.1 ∼0.25 STD (STD is the variance of the sequence), the sample entropy is rarely affected by $m$ and $r$. Therefore, $m$ was set to 2 and $r$ was set to 0.2 STD in this study.

In VMD decomposition of a time series, the components of different scales need to be separated so they occupy their own spectrum bandwidth, and the random noise in the time series needs to be distinguished from the modal components. Therefore, in this study, the average sample entropy and the cross-correlation coefficient $\rho_c$ were used to jointly determine the number of modes.

## HYBRID MODEL FOR FORECASTING

### GRU network

The GRU network is a type of recurrent neural network (RNN; *Cho et al., 2014*) that has been proposed to solve the problems of long-term memory in back propagation. The GRU network performs in a similar way to the LSTM network but is computationally cheaper. The structure of a GRU network is shown in Fig. 1 where $x_t$ is the input at the current node $t$, $h_{t-1}$ is the hidden state of transmission at the previous node $t$-1, $y_t$ is the output, and $h_t$ is the hidden state at $t$.

The GRU network has two gate states, as shown in Fig. 2. Here, $r$ and $z$ are the reset gate and the update gate, respectively, and can be written as:

$$r = \sigma\left(W_r\left[h_{t-1}, x_t\right] + b_r\right) \tag{16}$$

$$z = \sigma\left(W_z\left[h_{t-1}, x_t\right] + b_z\right) \tag{17}$$

where $\sigma$ is the sigmoid function which constrains the data in the interval [0,1]; $W$ and $b$ are the weight and threshold of the networks.

According to the reset gate $r$ and the hidden state of $h_{t-1}$, the reset signal can be obtained, as follows:

$$h'_{t-1} = h_{t-1} \otimes r \tag{18}$$

where $\otimes$ is the Hamiltonian operator.

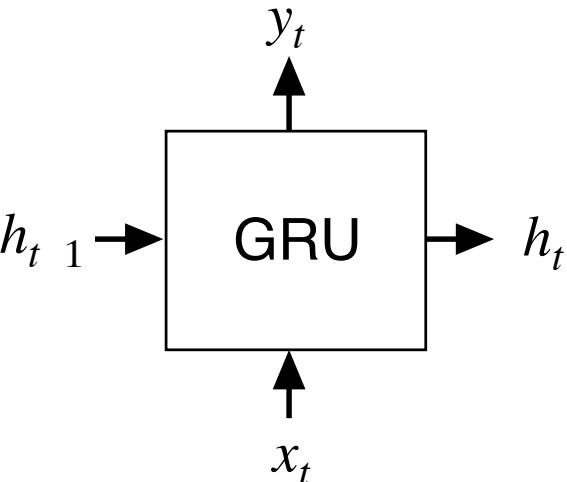

**Figure 1** **Input and output structure of the GRU network.**

The transmission state $h_t'$ is written as:

$$h_t' = \tanh\left(W_g\left[h_{t-1}', x_t\right] + b_g\right). \tag{19}$$

Lastly, the transmission state of $h_t$ can be written as:

$$h_t = (1-z) \otimes h_{t-1} + z \otimes h'. \tag{20}$$

## The improved optimization algorithm

The Adam algorithm updates weights by calculating the first and second moments of the gradient, improving the slow convergence problem caused by the fixed learning rate in the gradient descent method. The Adam algorithm was used on the random adjustment parameters in this study to effectively improve the convergence rate.

Firstly, initializing the learning rate $\mu$, and using optimization parameters $W_t$, the gradient $g_t$, the first moment $m_t$ of the gradient and the second moment $v_t$ of the gradient can be calculated, iteratively, as follows:

$$g_t = \nabla W_t f(W_t) \tag{21}$$

$$m_t = \beta_1 m_t + (1-\beta_1)g_t \tag{22}$$

$$v_t = \beta_2 v_{t-1} + (1-\beta_2)g_t^2 \tag{23}$$

where $\beta_1$ and $\beta_2$ are the decay rates of the first and the second moments, respectively.

Then, the deviations of the first and second moments of the gradient can be calculated as:

$$m_t' = m_t/(1-\beta_1) - \eta_1 g_t \tag{24}$$

$$v'_t = v_t/(1 - \beta_2) - \eta_2 g_t^2 \tag{25}$$

where $\eta_1$ and $\eta_2$ are the adjustment parameters of the first and second moments and the random numbers on interval [0,1], respectively.

Lastly, the parameters can be updated using the following formula:

$$w_t = w_{t-1} - \mu m'_t / \left( \sqrt{v'_t} + \varepsilon \right) \tag{26}$$

where $\varepsilon$ is the allowable error to prevent a zero value in the iterative process.

### Forecasting process

In this work, a hybrid model of adaptive VMD and the GRU network is proposed to reduce frequency aliasing and eliminate the randomness of the load series. The electrical load series is represented as $\{x(1), x(2), \ldots, x(N)\}$, where $N$ is the number of samples. When $C_i$ is the decomposed mode, the prototype mode is calculated, as follows:

$$C_i = \{c_i(1), c_i(2), \ldots, c_i(N)\}, i = 1, 2, \ldots, M. \tag{27}$$

The forecasting value of the prototype mode at time $(N+1)$ is $\hat{c}_i(N+1)$ using the GRU networks. Reconstructing other components after removing the residual sequence, the forecasting result at time $N+1$ is calculated, as follows:

$$\hat{x}(N+1) = \sum_{i=2}^{M} \hat{c}_i(N+1). \tag{28}$$

Figure 3 illustrates the load forecasting process of the hybrid model. First, the load series is decomposed using the adaptive VMD model. Then, each mode is forecasted in the next time using the improved GRU model. Finally, the sum of the IMF is the forecasting result of the load series. The forecasting process is as follows:

(a) The IMF of each modal component is obtained by initializing the number of modes and decomposing the load data using the adaptive VMD model.

(b) The cross-correlation coefficient $\rho_c$ between the noise residue and IMF is calculated, using Eq. (11).

(c) The sample entropy of each mode and the modal residue, and the average sample entropy (ASE) are calculated using Eqs. (9) to (15) under different numbers of modes.

(d) The decomposition number $K$ is selected, corresponding to the minimum of ASE and $\rho_c$.

(e) The modal components at the next time point are then forecasted using the improved GRU network.

(f) The final result is obtained by reconstructing the modal forecasting components.

## RESULTS

To analyze the forecasting performance, the root mean square error ($RMSE$), the mean absolute error ($MAE$) and the mean absolute percentage error ($MAPE$) were calculated, as

$$r = \quad \left( \boxed{W_r} \; \begin{array}{c} \boxed{x_t} \\ \boxed{h_{t-1}} \end{array} \right)$$

$$z = \quad \left( \boxed{W_z} \; \begin{array}{c} \boxed{x_t} \\ \boxed{h_{t-1}} \end{array} \right)$$

**Figure 2  Gate structure of the GRU network.**

follows:

$$RMSE = \sqrt{\frac{\sum_{n=1}^{N}(p_n - \hat{p}_n)^2}{N}} \tag{29}$$

$$MAE = \frac{\sum_{n=1}^{N}\left|p_n - \hat{p}_n\right|}{N} \tag{30}$$

$$MAPE = \frac{\sum_{n=1}^{N}\left|\frac{p_n - \hat{p}_n}{p_n}\right|}{N} \times 100\% \tag{31}$$

where $p_n$ and $\hat{p}_n$ represent the actual value and the forecasted value at the time $n$, respectively, and $N$ is the number of samples.

As a case study, experimental electrical data were taken from the Shaanxi province, China. The original load series was a one-minute interval, with data extracted every 15 min to form the data set. Because of the weekly periodicity of the load series, the final set of data included 672 samples.

## Results of the adaptive VMD model

The ASE of each component, except the residual and cross-correlation coefficient $\rho_c$, were calculated, and then the minimum values of the mode number $K$ were identified

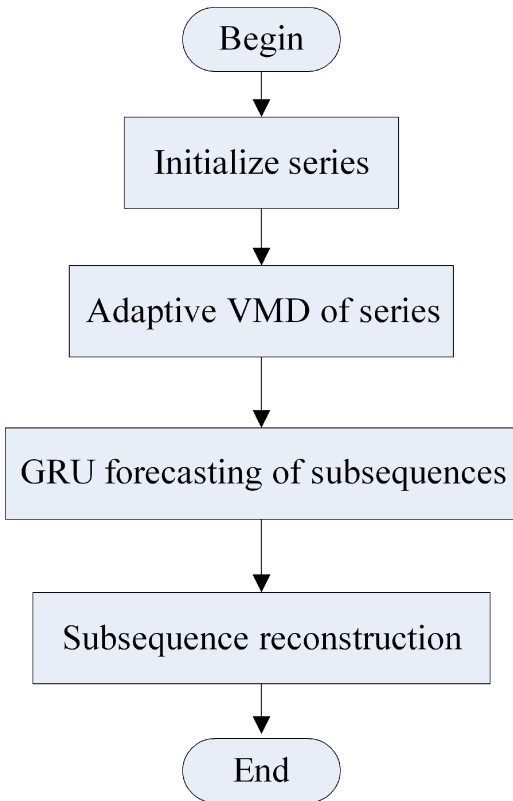

**Figure 3** Forecasting process of the hybrid model.

corresponding to the minimum *ASE* and $\rho_c$. The minimum *ASE* indicates that the similarity of each IMF was high, and that the sequence was more "orderly." The minimum $\rho_c$ means that the correlation between the residual sequence and the modal sequence was the smallest, meaning the reconstructed mode, except the residual, is closer to the real load sequence. Figure 4 shows the ASE and the cross-correlation coefficient $\rho_c$ at different time points; ASE reached the minimum value when the number of decompositions was 5, and $\rho_c$ was the smallest when $K = 5$.

The proposed adaptive VMD model decomposes the original electrical load series. Figure 5 shows the decomposition results of the load series of a day, consisting of 96 samples. In this figure, mode IMF4 has a small amplitude and violent random noise, giving it a residual sequence. The frequencies of the other modes decrease from top to bottom, revealing the short-term and long-term characteristics of the load series.

## Results of the improved GRU network

A training set was created from 100 groups of load series, with the other load series used as the test set. The default values of related parameters are given in Table 1.

The forecasting performance of the improved GRU network was compared with the original GRU model, with both methods using the same training and test data. Table 2 shows the comparison results of both forecasting error and time. Each error index

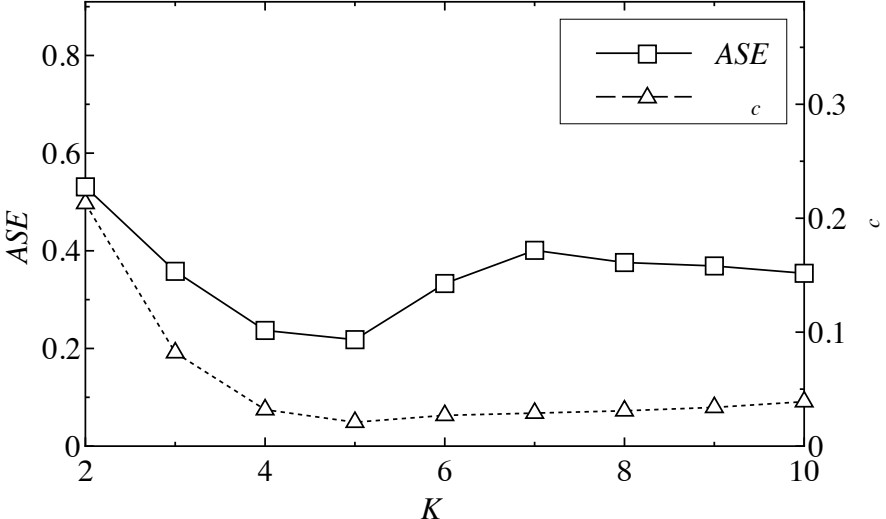

**Figure 4** ASE and $\rho_c$ of a load series of different $K$.

was based on the forecasting value of 672 datapoints. There were 100 groups of error index values in total, and the average value was the error result shown in Table 2. The forecasting error value of the improved GRU model proposed in this article was less than the forecasting error value of the original GRU model. The forecasting time of the improved GRU model was also significantly reduced compared to the original GRU model because the Adam algorithm uses a random adjustment of parameters.

Table 3 shows the forecasting error for different numbers of modes. When the number of modes was four, the forecasting error was the smallest, verifying the reliability of the adaptive VMD model.

## Results of the hybrid model

The proposed forecasting model was then verified using different types of measured data. Figure 6A and 7A show the actual valuables and forecasting results (original GRU model and our hybrid model) on a working day and a non-working day, respectively. In Fig. 6A, the RMSE of the original GRU model was 335.7 MW and the RMSE of the proposed model was 334.5 MW. In Fig. 7A, the RMSE of the original GRU model and the proposed model were 335.9 MW and 334.6 MW, respectively. Since these differences were small, Figs. 6B and 7B further illustrate the comparisons with smaller units on the load ($y$-axis) and less forecasting points ($x$-axis). These figures show that the proposed hybrid model had better forecasting performance than the original GRU model.

The number of hidden neurons is an important parameter that can affect forecasting performance. Table 4 shows the forecasting results of different numbers of neurons of the hidden layer, with all other parameters being optimal. The forecasting error was the smallest when the number of neurons was 40, but the optimal number of neurons was different with different datasets.

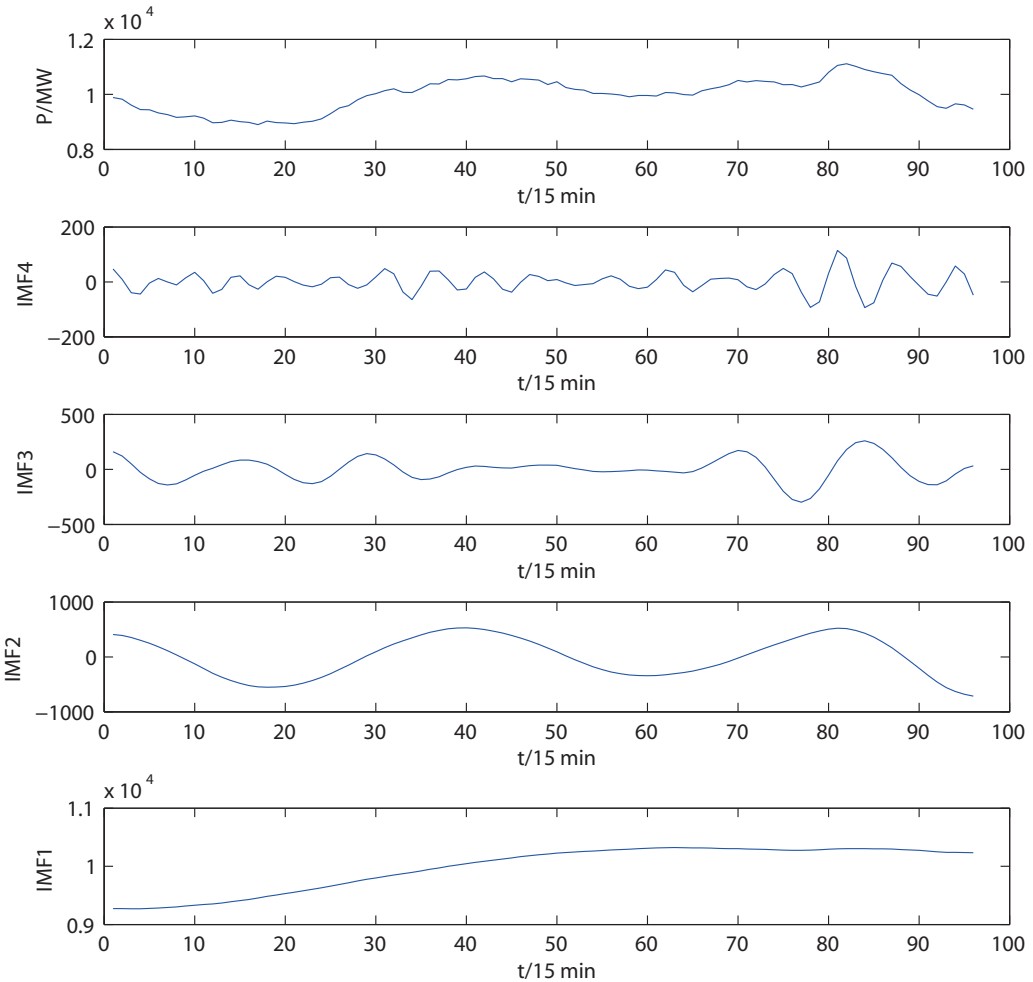

**Figure 5** Electrical load series and decomposition results of the adaptive VMD.

**Table 1 Default parameter settings of the improved GRU network.**

| Parameters | Default valuables |
| --- | --- |
| Number of hidden layers | 2 |
| Number of Samples | 100 |
| Learning rate | 0.01 |
| Number of neurons in hidden layer 1 | 50 |
| Number of neurons in hidden layer 2 | 50 |
| Number of input sequences | 672 |
| Momentum parameter | 0.5 |
| Maximum number of iterations | 2500 |
| Optimization algorithm | Improved Adam algorithm |

**Table 2** Comparison of forecasting performance of different GRU models.

| Methods | RMSE (MW) | MAE (MW) | MAPE (%) | Training time (s) | Forecasting time (s) |
|---|---|---|---|---|---|
| GRU | 336 | 201 | 1.931 | 323 | 3.59 |
| Improved GRU | 334 | 199 | 1.924 | 246 | 1.41 |

**Table 3** Forecasting error under different numbers of modes.

| Number of modes K | RMSE (MW) | MAE (MW) | MAPE (%) |
|---|---|---|---|
| 2 | 347 | 217 | 2.103 |
| 3 | 340 | 214 | 2.012 |
| 4 | 334 | 199 | 1.924 |
| 5 | 337 | 210 | 2.006 |
| 6 | 342 | 220 | 2.118 |
| 7 | 347 | 218 | 2.095 |

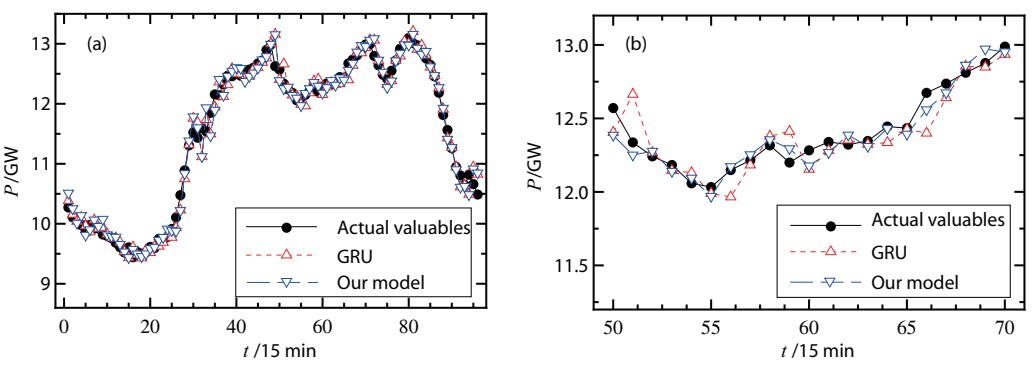

**Figure 6** (A–B) Forecast results of power loads on a working day.

The proposed hybrid forecasting model was also compared with the following classical statistical models: the ARIMA model, support vector regression (SVR), machine learning (Elman neural network), and the combined model. The parameter settings of the ARIMA model were set based on *Lee & Ko (2021)*, and the selection of model order was based on the AIC criterion. The kernel function of the SVR model was the Gaussian radial basis function and the kernel parameters were optimized based on *Sina & Kaur (2020)*. The parameter settings of the Elman neural network were set according to *Xie et al. (2020)*, the optimization algorithm adopted the traditional gradient descent method, and the number of neurons in both hidden layers was 40. The single method selection and parameter settings of the combined model were based on *Li & Chang (2018)*.

Table 5 illustrates the comparison of forecasting error of the above methods. Compared with the traditional statistical model and machine learning, the hybrid model proposed in this work had a higher forecasting accuracy. The traditional statistical

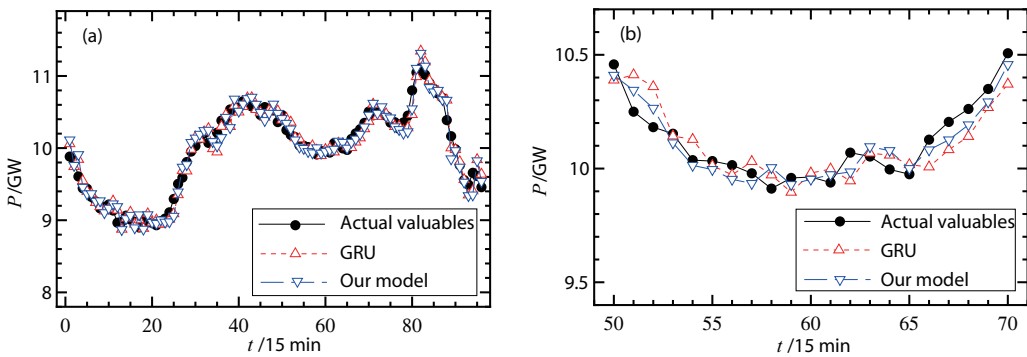

**Figure 7** (A–B) Forecast results of power loads on a non-working day.

**Table 4** Forecasting error under different numbers of neurons.

| Number of neurons in hidden layer 1 | Number of neurons in hidden layer 2 | RMSE (MW) | MAE (MW) | MAPE (%) |
|---|---|---|---|---|
| 10 | 10 | 359 | 208 | 2.101 |
| 20 | 20 | 354 | 205 | 1.999 |
| 30 | 30 | 338 | 197 | 1.912 |
| 40 | 40 | 332 | 194 | 1.887 |
| 50 | 50 | 339 | 198 | 1.986 |
| 60 | 60 | 345 | 203 | 2.002 |

learning method only obtained the evolution characteristics of the time series based on a limited number of samples, making it difficult to forecast long-term evolution and reversal characteristics of a time series. The machine learning methods, such as the radial basis function (RBF) and back-propagation (BP) neural network, have poor forecasting ability of a time series, and the Elman network is unable to forecast the long-term dependence of a time series. The proposed hybrid model, based on the GRU network, is a deep learning method, which obtains the evolution characteristics of sequences based on a large number of data, so the forecasting accuracy is higher. Because it is a deep learning method, the training time of the hybrid model based on the GRU network is much longer than that of the traditional statistical model and machine learning method.

Finally, the forecasting performance of the hybrid model in this work was compared with three state of the art methods: the LSTM network, the GRU network and the QRNN model. The LSTM algorithm and parameter settings were based on *Rafi et al. (2021)*. The selection and parameter value of the GRU network were based on *Shen et al. (2021)*. The setting and parameter values of the QRNN network were based on *Cannon (2011)*. Table 6 compares the forecasting performance of the above methods. The average forecasting error value shows that, compared with the LSTM forecasting method combined with EMD and VMD, the AVMD model proposed in this article decomposed the sequence more accurately and improved the accuracy of the GRU network forecasting results. The maximum and minimum errors also verified the stability and reliability of

**Table 5  Comparison of forecasting error of different forecasting methods.**

| Forecasting error | | Forecasting method | | | | |
|---|---|---|---|---|---|---|
| | | ARIMA | SVR | Elman network | Combinational model | Our model |
| MAPE (%) | Averages | 3.697 | 3.432 | 3.789 | 3.218 | 1.887 |
| | Minimum | 1.824 | 1.896 | 1.743 | 1.182 | 1.176 |
| | Maximum | 4.719 | 4.645 | 4.803 | 4.410 | 2.875 |
| MAE (MW) | Averages | 326 | 297 | 359 | 281 | 194 |
| | Minimum | 189 | 196 | 181 | 167 | 165 |
| | Maximum | 510 | 468 | 514 | 449 | 262 |
| RMSE (MW) | Averages | 475 | 437 | 507 | 421 | 332 |
| | Minimum | 316 | 328 | 303 | 280 | 277 |
| | Maximum | 773 | 731 | 778 | 682 | 395 |

**Table 6  Comparison of performance of deep learning methods.**

| Forecasting error | | Forecasting method | | | | | |
|---|---|---|---|---|---|---|---|
| | | EMD-LSTM | VMD-LSTM | EMD-GRU | VMD-GRU | QRNN | Our model |
| MAPE (%) | Averages | 1.951 | 1.937 | 1.928 | 1.904 | 1.910 | 1.887 |
| | Minimum | 1.236 | 1.230 | 1.242 | 1.198 | 1.203 | 1.176 |
| | Maximum | 3.294 | 3.295 | 3.201 | 3.187 | 3.204 | 2.875 |
| MAE (MW) | Averages | 204 | 201 | 200 | 196 | 197 | 194 |
| | Minimum | 179 | 177 | 183 | 169 | 180 | 165 |
| | Maximum | 287 | 290 | 284 | 271 | 279 | 262 |
| RMSE (MW) | Averages | 348 | 342 | 340 | 335 | 338 | 332 |
| | Minimum | 291 | 288 | 299 | 280 | 283 | 277 |
| | Maximum | 413 | 418 | 408 | 397 | 402 | 395 |
| Training time (s) | | – | 48.4 | 49.7 | 34.2 | 35.6 | 35.9 | 32.3 |

the hybrid model. Compared with other deep learning networks, the parallel structure of the hybrid model based on the GRU network significantly shortened the training time, making this model suitable for short-term power load forecasting.

Because it is a deep learning method, the hybrid model requires a large number of training samples, and therefore the training time is longer than that of machine learning methods. Training time could be further reduced by reducing the number of training samples and batch size through a consideration of the periodicity of the load series. Furthermore, based on actual data, a reasonable network structure could be optimized, such as the number of network layers and nodes, without significantly reducing forecasting accuracy.

## DISCUSSION

The results of this study show the effectiveness of the established hybrid GRU network with adaptive VMD for forecasting electrical load. The number of modes in adaptive

VMD decomposition is determined using the average sample entropy and the cross-correlation coefficient, improving forecasting performance. The adaptive VMD decomposition eliminated the randomness of the series, and better reflected the time scale characteristics of every subsequence, improving load forecasting performance, although it increased the training time of the model.

Furthermore, we clarify the research gaps filled in by the proposed forecasting model. Because load series have multiple periods and are nonlinear, the proposed model can improve forecasting accuracy by decomposing the load series into multiple sub-sequences. It is also difficult to determine the number of modes in VMD, and the proposed adaptive VMD can both determine the number of modes and improve the reliability of decomposition using average sample entropy and the cross-correlation coefficient. Finally, the proposed hybrid model improves forecasting accuracy at the cost of increased computational time.

The computational complexity of the proposed hybrid model is similar to that of LSTM, GRU, and RNN. Although the hybrid model has a longer computational time than traditional machine learning methods, its forecasting accuracy is significantly improved. The size of the network and the number of training samples could be further reduced in practical applications based on data characteristics, reducing computational complexity and the corresponding computational time, making the hybrid model appropriate for practical short-term power load forecasting.

## CONCLUSIONS

This study established a hybrid model of adaptive VMD and the GRU network and applied the model to short-term electrical load forecasting. The developed adaptive VMD method determines the modal number using average sample entropy and mutual correlation. The developed GRU network reduces training time by adding the random adjustment parameters to the Adam algorithm. The hybrid model reduces frequency aliasing and the randomness of the series, so its forecasted loads are close to the actual load data.

Some statistical models and machine learning methods, including ARIMA, SVR, the Elman networks, and the combined model, and some state of the art models including the LSTM method and the QRNN model, were compared with our proposed hybrid model. The values of MAPE, MAE and RMSE were reduced in comparison with the traditional statistical models. The training time of the hybrid model was much smaller than that of the deep learning method, and the proposed hybrid model had better performance in short-term load forecasting.

### Funding

This work was supported by the National Natural Science Foundation of China (62003259). The funders had no role in study design, data collection and analysis, decision to publish, or preparation of the manuscript.

### Grant Disclosures

The following grant information was disclosed by the authors:
the National Natural Science Foundation of China: 62003259.

### Competing Interests

The authors declare that there are no competing interests.

### Author Contributions

- Chun-Hua Wang conceived and designed the experiments, performed the experiments, analyzed the data, performed the computation work, prepared figures and/or tables, authored or reviewed drafts of the article, and approved the final draft.
- Wei-Qin Li conceived and designed the experiments, analyzed the data, authored or reviewed drafts of the article, and approved the final draft.

### Data Deposition

The code and raw data are available in the Supplemental Files.

### Supplemental Information

Supplemental information for this article can be found online at http://dx.doi.org/10.7717/peerj-cs.1514#supplemental-information.

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
