# Peer review of "A hybrid model of modal decomposition and gated recurrent units for short-term load forecasting"

_PeerJ Computer Science, doi:10.7717/peerj-cs.1514_

## Round 0.1 · original submission · Minor Revisions

The authors need to address the reviewers' comments particularly grammatical and typo mistakes. They need to provide additional proof to validate the work and to justify their contribution to the body of knowledge.

·

Basic reporting

1. Section Variational Mode Decomposition (VMD): Hilbert transform equation is used. Its reference should be cited.
2. A revision of the draft is required to address small grammatical mistakes throughout the text.
3. Table 3 and table 4 have same caption. The captions and the related text in the discussion in manuscript need to clarify the difference between both tables.
4. Table 5: font is not uniform.
5. References are outdated. Literature and supporting citations should be recent.

Experimental design

1. The manuscript requires a little bit more clarification on the research gap. What exactly was the research gap that their hybrid model attempts to fill and what is the cost of it?

Validity of the findings

1. Fig 6 and 7 show the authors' hybrid model vs. GRU alone. However, the improvement is not clearly visible. Since the difference is in fairly smaller units, a separate graph should be included highlighting the improvement as compared to GRU alone, with smaller units on the y-axis.
2. The authors commented about the training time of the model, however they did not suggest any measures that could be taken in their future works to compensate for this increased training time.
3. No comparison/ validation has been made for computational complexity of the proposed hybrid algorithm. The success of combining 2 techniques is obvious; however, the cost of doing this should also be discussed to justify whether it is practically usable or not.

Additional comments

Overall, the manuscript is well-written minus few grammatical mistakes. However, more proof is required on the validation of the work to justify its contribution to the body of knowledge.

·

Basic reporting

In this article, the authors design a hybrid gated recurrent unit (GRU) network to predict electrical load forecasting at short time scales. The language of the paper is clear and ambiguous except for a couple errors in the text that I detail below:

1. Line 58-59: "Then as improved GRU network and a hybrid model are introduced ..."-> "Then *an* improved GRU network and a hybrid model are introduced ...."

2. Line 100: "beis" -> "be"

Experimental design

The experiment design are interesting but there are some claims made in this paper that are disputable. I am still saying " Minor revisions" because those disputable claims can be fixed by addressing the language used for their descriptions as well as with edits to the figures.

Major comments:

1. Figures 6 and 7 is almost unreadable given the number of points in the time-series, and the chosen markers. It is hard to determine how the predictions align with the true values. I suggest the authors do one of the following two options to alleviate this issue - subsample the number of points on the time-series to show clearly how the predictions align with the true values, use different markers such as circles and dots '.' so that the dots overlap with the circles. Additionally, the authors can report overall error such as root-mean-squared-error (RMSE) in the two figures to get a general sense of how their models perform

2. Lines 224-225: The authors claim that average sample entropy (ASE) reaches it min at K=4, however from Fig 4, it's clear that this min at K=5. Moreover, the min value of p_c is also at K=5 (or K>=5) unlike at K=4 as is written there

3. Line 93: Please provide a citation for spectrum breakage. It is a crucial piece in the experiment design and is not very widely known why spectrum breakage would occur.

4. Line 88: The Lagrange equation used for the solution calculation should be provided. Also, please describe in some detail how is this solution obtained?

5. [Minor] Line 269: Please expand RBF and BP neural network. What does BP stand for?

Validity of the findings

The results obtained in this experiment are valid but I believe they could be represented better. Please refer to my comment 1 in the previous section "Experimental design" on an important suggestion to make the report better

---

## Round 0.2 · accepted · Accept

The manuscript has been revised and improved based on the comments given by the reviewers.